# Mechanistic Insights into Hepatic Pathogenesis of Dengue Virus Serotype-2: Host–Virus Interactions, Immune Dysregulation, and Therapeutic Perspectives

**DOI:** 10.3390/ijms262210904

**Published:** 2025-11-10

**Authors:** Tharshni Naidu A. Rama Ravo, Wei Boon Yap

**Affiliations:** 1Biomedical Science Programme, Faculty of Health Sciences, Universiti Kebangsaan Malaysia, Kuala Lumpur 50300, Malaysia; tharshninaidu28@gmail.com; 2Center for Toxicology and Health Risk Studies, Faculty of Health Sciences, Universiti Kebangsaan Malaysia, Kuala Lumpur 50300, Malaysia; 3One Health UKM, Universiti Kebangsaan Malaysia, Bangi 43600, Selangor, Malaysia

**Keywords:** dengue virus serotype 2, liver pathogenesis, hepatocytes, cytokine storm

## Abstract

Dengue virus serotype 2 (DENV-2) is a predominant cause of severe dengue and a key determinant of dengue-associated liver injury. This review integrates recent findings on the molecular and cellular mechanisms of DENV-2 hepatotropism, focusing on viral replication, cellular stress responses, and immune-mediated damage. The interplay between hepatocytes, Kupffer cells, and innate and adaptive immune responses, culminating in cytokine storm and immune-mediated hepatocyte apoptosis, is dissected. Integrating in vitro and in vivo findings, this review highlights how viral replication and immune dysregulation converge to cause hepatic injury. Future research should prioritize antiviral, immunomodulatory, and hepatoprotective approaches aimed at reducing the risk of dengue-associated liver failure.

## 1. Introduction

Dengue virus (DENV), a mosquito-borne flavivirus, causes nearly 390 million infections annually [1]. The four DENV serotypes, namely DENV-1, DENV-2, DENV-3, and DENV-4, pose varying risks of disease severity. These severe disease manifestations resulted from exaggerated immune responses, leading to increased vascular permeability, hemorrhage, and multiorgan failure [2]. Among these, DENV-2 is particularly associated with more severe manifestations, such as dengue hemorrhagic fever (DHF) and dengue shock syndrome (DSS), especially in secondary dengue [1,2]. Moreover, DENV-2 has been implicated in recurrent outbreaks and the global escalation of dengue incidence, as well as elevated hospitalization and mortality rates, highlighting its importance for continued investigation [3,4]

Liver dysfunction is a key feature of severe dengue, often presented as elevated liver enzymes, hepatomegaly, jaundice, and, in extreme cases, acute liver failure. As a primary site for DENV replication and immune modulation, hepatocytes exhibit cytopathic effects (CPEs), including cell rounding, detachment, and death upon DENV-2 infections. The elevated aspartate transaminase (AST) and alanine transaminase (ALT) levels correlate directly with disease severity, reflecting direct viral injury, immune-mediated damage, and oxidative stress [5]. Recent reviews have examined dengue pathogenesis from immunological, hepatic, and histopathological perspectives [6,7,8]; however, the molecular mechanisms underlying DENV-2-induced liver injury remain poorly defined. Despite its clinical importance, the precise pathways linking viral replication, immune dysregulation, and hepatocellular damage are not fully understood. Therefore, this review aims to elucidate the mechanisms of DENV-2 infection in hepatocytes, the resulting immune-mediated liver injury, and the associated clinical outcomes.

## 2. Pathophysiology of DENV-2 Infection in Liver

The liver plays a key role in DENV-2 infections, acting as a viral replication site and a mediator of systemic inflammation. Biochemical markers such as ALT and AST not only reflect hepatocellular injury but also correlate with increased disease severity and extended hospital stays [9]. Following viral entry into hepatocytes, DENV-2 undergoes replication through receptor-mediated endocytosis, RNA release, translation, replication on the endoplasmic reticulum (ER), and assembly of mature virions that are released via exocytosis (Figure 1) [10]. This replication cycle disrupts ER and mitochondrial function, leading to oxidative stress and apoptosis, which exacerbate hepatocellular injury.

Generally, DENV-2 infects hepatocytes, which then triggers stress on the ER and mitochondria [11]. ER stress disrupts cellular protein synthesis and calcium homeostasis, while mitochondrial dysfunctions stimulate the synthesis of reactive oxygen species (ROS), leading to oxidative damage, apoptosis, and hepatocyte death [12]. Additionally, it triggers a strong innate immune response, leading to a cytokine storm marked by the elevated levels of tumor necrosis factor-alpha (TNF-α), interleukin-6 (IL-6), and interferon-gamma (IFN-γ), which exacerbate liver damage. Clinical findings have linked the increase in those cytokines to the severity of dengue-related liver dysfunctions [13]. In the DENV-2-infected liver, activated Kupffer cells further amplify inflammation by releasing ROS and pro-inflammatory cytokines. These phenomena signify the key involvement of the liver in severe dengue and highlight the need for more targeted therapeutic strategies.

### 2.1. Histopathological Findings in DENV-2-Induced Liver Damage

Histological studies involving liver biopsies of DENV-infected patients revealed a spectrum of pathological changes, indicating hepatocyte injury and inflammation.

#### 2.1.1. Hepatocyte Apoptosis and Necrosis

Hepatocyte apoptosis is a key feature of DENV-2 infections in the liver, driven by virus-induced stress. In DENV-2 infections, the accumulated ROS and cytokines cause oxidative damage, ER stress and mitochondrial dysfunction [6]. When present in excess, ROS can induce lipid peroxidation and protein oxidation, damaging mitochondrial membranes and worsening hepatocyte injury through apoptosis and necrosis [14]. This also explains the elevated ALT and AST levels in the plasma due to leakage of cellular contents into the bloodstream. The oxidative damage is worsened by ER stress and mitochondrial dysfunction, followed by the release of pro-inflammatory cytokines. Infiltration of immune cells subsequently ensues, further exacerbating hepatocyte injury [15]. The cellular damage and inflammation eventually cause infected hepatocytes to exhibit morphological changes such as cell shrinkage, chromatin condensation, and membrane blebbing, leading to apoptotic body formation and clearance by immune cells [16].

#### 2.1.2. Steatosis (Fatty Liver)

Steatosis, marked by the accumulation of lipid droplets in hepatocytes, is another key pathological feature of DENV-2 infections in the liver. DENV-2 replication disrupts fatty acid metabolism, causing lipid buildup that impairs liver functions and increases oxidative stress [17]. As a result, the normal liver functions are interrupted alongside the promotion of cellular inflammation.

Oxidative stress and inflammation caused by excessive lipid accumulation in hepatocytes can occur via multiple interconnected mechanisms. Generally, excessive free fatty acids (FFAs) and triglycerides undergo lipotoxicity, which progressively leads to mitochondrial dysfunction and increased ROS production [17]. When mitochondrial β-oxidation becomes overwhelmed, excess FFAs are subjected to peroxisomal and microsomal oxidations, which inadvertently generate excessive ROS as byproducts. Altogether, the ROS level is tremendously elevated, which in turn affects the functions and integrity of cellular components, including lipids, proteins, and DNA [18]. In a nutshell, steatosis exacerbates DENV-2-induced liver injury and is associated with greater disease severity.

#### 2.1.3. Mononuclear Cell Infiltration

Studies have shown that DENV-2 infections lead to a significant infiltration of mononuclear immune cells, including monocytes, macrophages, lymphocytes, and neutrophils, in the liver [19]. These immune cells initially are activated to control the virus infection; their activation, however, results in excessive release of cytokines and inflammatory mediators that cause collateral liver tissue damage and injury [20]. Monocytes and macrophages produce high levels of TNF-α and interleukins, promoting hepatocyte apoptosis and fibrosis [21]. Neutrophils are keen to induce oxidative stress through ROS production, damaging hepatocyte membranes and disrupting liver architecture [22]. Lymphocytes, particularly cytotoxic T cells, can induce direct hepatocyte injury through perforin and granzyme-mediated mechanisms, further compounding liver dysfunction [23]. Additionally, hyperactivated Kupffer cells liberate inflammatory mediators, increasing vascular permeability. It is, therefore, essential to highlight the impact of infiltrated mononuclear cells on endothelial dysfunction and microvascular disturbances, as these adversities can disintegrate liver functions [24].

#### 2.1.4. Kupffer Cell Hyperplasia and Sinusoidal Congestion

During DENV-2 infections, activated Kupffer cells undergo hyperplasia, leading to uncontrollable inflammation [13]. This is mainly due to the release of pro-inflammatory cytokines and mediators such as IL-1β and TNF-α, along with high levels of ROS and nitric oxide (NO) [25]. Besides cellular inflammation, the high levels of pro-inflammatory cytokines can also upregulate the expression of adhesion molecules on sinusoidal endothelial cells, thereby increasing leukocyte infiltration and causing sinusoidal congestion [26]. Accumulatively, Kupffer cell hyperplasia promotes sinusoidal congestion that further amplifies localized inflammation and hepatic injury [27]. Moreover, Kupffer cell activation also enhances phagocytosis of platelets and red blood cells, contributing to thrombocytopenia and worsening liver microvascular disturbances [28]. These changes promote microvascular thrombosis, blood pooling within sinusoids, and elevated intrahepatic pressure, which collectively reduce oxygen and nutrient delivery to hepatocytes [29]. As oxygen exchange depletes and detoxification functions are disrupted, the overall liver function deteriorates [30].

## 3. Pathogenesis of DENV-2 in the Liver

### 3.1. Viral Entry

In the liver, DENV-2 utilizes primary receptors for initial host–virus interactions and co-receptors to enhance binding and internalization. These receptors, expressed on Kupffer cells, dendritic cells, endothelial cells, and hepatocytes, play distinct roles in the pathogenesis of DENV-2. Table 1 summarizes the key receptors and co-receptors responsible for DENV-2 entry into liver cells.

In brief, DC-SIGN functions as a primary receptor and binds directly to the DENV-2 E protein, facilitating viral attachment to Kupffer and dendritic cells [31]. The MR acts as a co-receptor, providing additional adhesion through glycan recognition for full viral entry [32]. AXL, another co-receptor, recognizes phosphatidylserine on the viral envelope via Gas6, an entry mediator. These receptors interact cooperatively to direct DENV-2 dissemination within the liver, contributing to the pathogenesis of DENV-2 in the liver [33].

#### 3.1.1. DC-SIGN

DC-SIGN is a C-type lectin receptor predominantly expressed on dendritic cells, which are crucial in orchestrating immune response [31]. It binds to high-mannose glycans on the DENV envelope (E) protein, facilitating viral attachment and entry into host cells. Subsequently, viral spread is promoted [34]. DENV-2 can hijack DG-SIGN-mediated immune signaling to suppress cellular antiviral responses, hence immune evasion by the virus. This usually occurs via interference with type I interferon (IFN) production, especially by altering Toll-like receptor (TLR) and RIG-I-like receptor (RLR) pathways, thereby dampening the antiviral state of infected and neighboring cells [35].

Additionally, activation of DC-SIGN-related pathways by DENV has been shown to enhance IL-10 production, a cytokine known for its immunosuppressive effects [36]. This immunoregulatory environment may slow down antiviral actions, allowing viral persistence. Owing to the immunoregulatory roles of DC-SIGN in dengue pathogenesis and as a key receptor in the early stage of DENV-2 infections, it has been recommended as a potential target for therapeutic interventions aimed at blocking DENV entry.

#### 3.1.2. MR

The MR, expressed on both macrophages and dendritic cells, plays a critical role in recognizing carbohydrate structures on the surface of DENV, particularly the viral E glycoproteins. It facilitates viral attachment and entry into host cells, hence promoting viral replication [32,37]. In the liver, MR is mostly responsible for DENV-2 infections in macrophages and other immune cells within the liver. Infected macrophages and dendritic cells serve as mobile reservoirs, carrying the virus through the lymphatic system and into the bloodstream [26]. During the migration to the secondary lymphoid organs and other tissues, the virus disseminates and causes viremia, amplifying the virus infection in multiple organ systems [38].

MR activation stimulates both local and systemic immune responses. The sustained stimulation of immune responses in hepatic tissue exacerbates inflammation, contributing to hepatocellular injury [39]. Thus, MR not only supports viral spread within the liver but also facilitates systemic dissemination and immune dysregulation, contributing to hemorrhagic complications.

#### 3.1.3. AXL Receptor Tyrosine Kinase

Axl is a receptor tyrosine kinase of the TAM family expressed on various cell types, including hepatocytes and immune cells [40]. Axl has been identified as a key receptor in facilitating DENV-2 binding and internalization, particularly in liver cells via clathrin-mediated endocytosis [41]. Upon viral internalization, Axl-mediated signaling pathways are activated, regulating both innate and adaptive immune pathways. For example, in dendritic cells and macrophages, Axl-mediated signaling promotes the release of immunoregulatory cytokines, such as IL-10, which can dampen antiviral immunity [26]. Moreover, it also causes the influx and activation of immune cells into the liver, thus exacerbating liver inflammation and tissue damage [42]. In this light, the immunosuppressive milieu supports persistent viral replication, resulting in more severe clinical outcomes, such as hepatic dysfunction and increased viral load.

### 3.2. Dysfunctional Cellular Processes upon DENV-2 Infection

DENV-2 exerts a range of direct effects on hepatocytes, resulting in a series of cellular stress and dysfunctional events (Figure 2). These effects primarily arise from the virus hijacking the host cell machinery for replication, which disrupts normal hepatocyte functions.

#### 3.2.1. ER Stress

DENV-2 replication within the ER results in the accumulation of misfolded and unfolded proteins, disrupting ER homeostasis and initiating ER stress. This stress affects chaperones such as glucose-regulated protein 78/binding immunoglobulin protein (GRP78/BiP) and activates stress sensors including protein kinase RNA-like endoplasmic reticulum kinase (PERK), inositol-requiring enzyme 1 (IRE1), and activating transcription factor 6 (ATF6) [43]. Under normal conditions, the unfolded protein response (UPR) restores ER balance by promoting protein folding, degrading misfolded proteins, and temporarily reducing protein synthesis [44]. However, sustained DENV-2 replication can chronically activate the UPR, shifting its role from adaptive to pro-apoptotic and contributing to hepatocellular damage [45,46]. Prolonged ER stress also impairs immune regulation and triggers hepatocyte apoptosis or necrosis [12,47]. In severe cases, unregulated ER stress combined with inflammatory signaling exacerbates liver injury.

#### 3.2.2. Oxidative Stress

During DENV-2 infection, the excessive generation and accumulation of ROS are frequently observed and closely linked to pronounced oxidative stress [25]. Elevated ROS levels inflict damage on cellular lipids, proteins, and DNA, leading to lipid peroxidation, protein misfolding, and genomic instability [48,49,50,51,52]. These molecular disturbances compromise hepatocyte integrity and trigger inflammatory cascades. Moreover, ROS activate key signaling pathways, including nuclear factor kappa-beta (NF-κB) and mitogen-activated protein kinase (MAPK), thereby enhancing the production of pro-inflammatory cytokines such as TNF-α, IL-6, and IL-1β [53,54,55]. In addition, ROS-induced cellular injury promotes the assembly of the NOD-like receptor family, pyrin domain containing 3 (NLRP3) inflammasome and subsequent caspase-1 activation, which further amplifies IL-1β secretion and inflammatory damage [56]. Collectively, these oxidative and inflammatory responses exacerbate ER stress and mitochondrial dysfunction, thereby accelerating hepatic injury in severe dengue.

#### 3.2.3. Mitochondrial Dysfunction

Mitochondrial dysfunction represents a critical downstream outcome of ER stress and oxidative stress during DENV-2 infection. Perturbation of mitochondrial membrane potential and the consequent reduction in adenosine triphosphate (ATP) synthesis disrupt vital cellular functions, including protein translation, ion homeostasis, and detoxification capacity, leading to metabolic derangements and aggravation of hepatic pathology [57,58,59,60]. Damaged mitochondria release pro-apoptotic molecules such as cytochrome c and second mitochondria-derived activator of caspases/Direct IAP-binding protein with low pI (Smac/DIABLO) into the cytosol, where they engage apoptotic protease activating factor 1 (Apaf-1) to form the apoptosome, thereby activating caspase-9 and downstream effector caspase-3 within the intrinsic apoptotic cascade [61,62,63,64]. Concurrently, extrinsic death receptor signaling, mediated by Fas–FasL interactions and TNF-related pathways, triggers caspase-8 activation and BH3-interacting domain death agonist (BID) truncation, establishing cross-talk between extrinsic and intrinsic apoptotic routes [65]. The convergence of these apoptotic mechanisms, amplified by direct viral cytopathic effects and immune-driven hepatotoxicity, culminates in extensive hepatocyte apoptosis and necrosis, driving liver dysfunction and failure in advanced stages of DENV-2 infection.

### 3.3. Immune-Mediated Liver Injury

The immune response targeting DENV-2 infections is a double-edged sword; whilst it essentially inhibits DENV-2 replication, it also induces liver injury, especially in severe dengue cases (Figure 3). The armed immune cells synthesize and liberate pro-inflammatory cytokines to initiate anti-DENV-2 response. CD8+ T lymphocytes target infected hepatocytes via the T-cell receptor/major histocompatibility complex class-I (TcR/MHC-I)-antigen and cause cell death via the release of perforin and granzymes. CD4+ T cells, on the other hand, are responsible for signaling pro-inflammatory cytokine responses, such as IL-6 and TNF-α. Accumulatively, these events promote the occurrence of cellular inflammation, worsening hepatocyte damage and liver injury.

#### 3.3.1. Cytokine Storm

One of the major contributors to liver damage and inflammation in severe dengue is the excessive production of pro-inflammatory cytokines, often referred to as cytokine storm. Key cytokines involved in this response include TNF-α, IL-6 and IFN-γ [6,66,67]. Pro-inflammatory cytokines induce vasodilation through binding to their respective receptors on endothelial cells, triggering intracellular signaling cascades such as the NF-κB and Janus kinase/signal transducer and activator of transcription (JAK-STAT) pathways [68]. In addition, the cytokine-receptor interactions also cause upregulation of adhesion molecules, e.g., intercellular adhesion molecule-1 (ICAM-1), vascular cell adhesion molecule-1 (VCAM-1) and E-selectin, that are responsible for extravasation of immune cells into the liver. Simultaneously, other inflammatory mediators, including chemokines and ROS, are actively synthesized [27,69]. As a result, pro-inflammatory response takes place, promoting vascular permeability and inflammation. The vascular lining becomes leaky and fails to retain plasma, hence hemorrhage and adverse liver injury.

A cytokine storm is particularly prominent in secondary DENV infections, where the immune response is more robust and often associated with severe dengue diseases, including vascular leakage and shock [70]. The heightened immune response is primarily driven by antibody-dependent enhancement (ADE), where pre-existing, non-neutralizing antibodies from a previous infection facilitate DENV entry into immune cells, leading to increased viral replication and stronger activation of monocytes, macrophages, and T cells [71]. These activated immune cells produce and liberate excessive pro-inflammatory cytokines, such as TNF-α, IL-6, and IFN-γ, amplifying pro-inflammatory response and endothelial dysfunction [72]. Memory T cells from the previous infection may also become hyperactivated, producing high levels of inflammatory mediators and exacerbating the consequences of cytokine storm.

#### 3.3.2. T-Cell-Mediated Injury

In addition to the cytokine response, CD8+ cytotoxic T cells play a key role in liver damage during DENV-2 infections. Given its cytotoxic nature, CD8+ T cells recognize viral antigens displayed on infected hepatocytes and induce cell death through the release of cytotoxic molecules such as perforin and granzymes [23,73]. Viral antigens are processed inside antigen-presenting cells (APCs) and displayed on MHC-I to attract CD8+ T cells. Through interactions with TcR, the antigen-MHC-I complex triggers T-cell proliferation, activation and their cytokine secretion [74]. Once activated, CD8+ T cells release an effector molecule, namely perforin, that binds to the infected cell membrane and causes pore formation. Subsequently, another effector molecule, granzyme, is translocated into the infected cells and induces apoptosis. In addition to the intracellular pathway, CD8+ T cells also engage the Fas–FasL pathway to promote programmed cell death in hepatocytes, hence extensive liver damage and immune-mediated pathology [75].

Apart from CD8+ T cells, CD4+ T-cell signaling is equally essential in activating cytokine storms during DENV-2 infections. APCs, such as dendritic cells and macrophages, process and present viral antigens on major histocompatibility complex class-II (MHC-II) molecules, which are later recognized by CD4+ T cells through their TcR [2], leading to their proliferation into various effector subsets, including T-helper (Th) 1 and Th17 cells [76]. Once activated, Th1 cells release pro-inflammatory cytokines such as IFN-γ and TNF-α to enhance macrophage activation and amplify immune-mediated tissue damage [77]. Th17 cells, on the other hand, produce IL-17 to recruit neutrophils to sustain inflammatory responses [78]. The combination of these immune responses concerts a cascade of hepatocyte damage, contributing to both direct viral effects and immune-mediated injury [79].

## 4. In Vitro Models for DENV-2 Infections in the Liver

Each in vitro model provides distinct mechanistic insights into DENV-2 infection, particularly regarding viral tropism, replication dynamics, and hepatocellular injury. These models collectively enable the dissection of molecular events such as viral entry, ER stress induction, oxidative damage, and immune-mediated cytotoxicity that drive hepatic pathogenesis. Thus, a comprehensive understanding of DENV-2–induced liver injury requires the integration of findings across multiple experimental systems, allowing for a more robust and mechanistically grounded interpretation of disease progression.

### 4.1. Chang Liver Cell Line

Oxidative stress is one of the most widely studied cellular mechanisms pertinent to DENV-2 infections in Chang liver cells [80]. DENV-2-induced oxidative damage not only disrupts essential cellular functions but also induces pro-inflammatory response, causing hepatocyte injury. In addition, mitochondrial dysfunction has also been documented in DENV-2-infected Chang liver cells, characterized by impaired mitochondrial membrane integrity, depletion of ATP, and initiation of cytochrome C-mediated apoptosis [81].

Furthermore, DENV-2-induced ER stress reportedly activates UPR in Chang liver cells due to the accumulation of misfolded viral and host proteins. UPR is generally known to restore cellular homeostasis; however, any prolonged activation can overwhelm the homeostasis system and cause cell apoptosis or necrosis [82]. Collectively, the Chang liver cell line sheds light on a better understanding of DENV-2 pathogenesis in the liver, thereby serving as an infection model for studying DENV-2-induced liver injury. However, the Chang liver cell line is not without limitations. Owing to their transformed nature, Chang liver cells may not fully mimic the natural physiology of primary hepatocytes. It is, therefore, important to interpret results generated with the Chang liver cell line alongside in vivo data for a more accurate representation of DENV-2 infections in human hosts.

### 4.2. HepG2 Cell Line

The HepG2 cell line, derived from human hepatocellular carcinoma, provides a more physiologically relevant model for studying DENV-2 pathogenesis than the Chang liver cell line. It closely resembles primary hepatocytes in terms of oxidative stress and pro-inflammatory response in viral infections [21]. In this light, the HepG2 cell line is a valuable tool for investigating interactions between DENV-2 and hepatocytes, such as immune modulation and cellular apoptosis [83]. In studying DENV-2-induced apoptosis in HepG2 cells, the intrinsic and extrinsic apoptosis pathways, as evidenced by the activation of caspases 4, 7, 8, and 9, along with mitochondrial membrane potential changes, have been deliberated [83,84]. For instance, upregulation of tumor necrosis factor-related apoptosis-inducing ligand (TRAIL) and its receptor death receptor 5/TNF-related apoptosis-inducing ligand receptor 2 (DR5/TRAIL-R2). The upregulation is tightly regulated by NF-κB and specificity protein 1 (Sp1) [85]. Upon cellular apoptosis, apoptotic bodies and necrotic cells release damage-associated molecular patterns (DAMPs), followed by the expression of pro-inflammatory cytokines [85]. The entire process is almost identical to that observed in severe dengue cases [86].

Nonetheless, the carcinogenic nature of HepG2 cells renders several limitations in DENV-2 studies. Their altered gene expressions, dysregulated signaling pathways, and abnormal metabolic processes may influence the cellular response to the virus infection [85]. For instance, although DENV-2-infected HepG2 cells exhibit apoptosis, they are unable to exhibit some apoptotic elements, which can potentially affect the extent of virus-induced programmed cell death [85]. Furthermore, epigenetic modifications and chromosomal abnormalities in carcinogenic liver cells may influence the differential gene expression patterns and, therefore, cannot fully represent those of normal liver cells [87]. Whilst considering the HepG2 cell line as an in vitro infection model of DENV-2 in the liver, it is noteworthy that cancer-related alterations must be taken into account when extrapolating the empirical findings.

### 4.3. Primary Hepatocytes

Primary hepatocytes, isolated from human or animal livers, represent the most physiologically relevant model for studying DENV-2 infections due to their ability to retain key hepatocyte functions, including metabolic pathways and immune response [88,89]. These cells closely mimic the in vivo conditions and can therefore provide valuable insights into how DENV-2 modulates hepatocyte activities, hence better comprehension of DENV-2 pathogenesis in the liver [90]. In this light, primary human hepatocytes (PHHs) are proposed as the gold standard for DENV-2 research, such as viral entry and replication in liver cells and development of targeted antiviral strategies [90].

Nonetheless, primary cell lines are known for their relatively short lifespan and might vary physiologically from donor to donor [90]. To address these challenges, hepatocyte-like cells (HLCs) derived from human pluripotent stem cells (hPSCs) and human mesenchymal stem cells (hMSCs) have emerged as more promising alternatives [90]. HLCs can support productive DENV-2 infections and exhibit key cellular responses, making them a reliable and sustainable research platform [89]. In addition to human cell-derived models, mouse hepatocyte cell lines, such as AML12, have been occasionally employed in investigating DENV-2-induced oxidative stress and apoptosis [91]. These models provide a consistent platform for studying viral replication dynamics, anti-DENV immune response, and the potential of therapeutic interventions.

## 5. Potential Therapeutic Interventions

A variety of therapeutic interventions have been explored to counteract the multifactorial mechanisms underlying DENV-2-associated hepatic pathology. These approaches broadly encompass antiviral agents targeting viral replication and entry, immunomodulatory therapies aimed at tempering cytokine-driven inflammation, and hepatoprotective measures designed to restore redox balance and mitochondrial integrity (Table 2). Collectively, these strategies highlight the translational potential of integrated antiviral, anti-inflammatory, and cytoprotective modalities in mitigating dengue-induced liver injury.

Individually, details of the respective interventional approaches are discussed in the following subsections.

### 5.1. Targeting Viral Replication

A promising approach for combating viral infections often involves inhibiting viral replication. Recently, the use of small inhibitor molecules exhibiting antiviral properties has gained much attention [101]. Similar strategies have been employed in developing anti-DENV inhibitors, particularly those targeting NS3 protease and NS5 polymerase. DENV NS3 protease is responsible for processing the viral polyprotein, while the NS5 polymerase is involved in the viral RNA replication [92]. Targeted inhibition of viral proteins may attenuate viral replication dynamics and consequently mitigate hepatocellular injury, particularly during the acute phase of DENV infection when replication efficiency and viral load reach their peak [93]. The notion was positively proven in preclinical studies involving in vitro and animal models, providing a foundation for clinical evaluation [93]. In addition to enzyme inhibitors, receptor antagonists that block viral entry are extensively studied for their antiviral potential [94]. However, this approach is challenging for dengue, as DENV can enter host cells through multiple cellular receptors. Investigating molecular inhibitors with identical or comparable mechanisms that block DENV entry into hepatocytes using various in vitro and in vivo models could help elucidate their mechanisms of action and potentially enhance their anti-DENV efficacy [94].

### 5.2. Reducing Immune-Mediated Injury

Immune-mediated liver damage, driven by cytokine storms and excessive inflammation, is often seen in severe DENV-2 cases. Targeting key pro-inflammatory cytokines, such as TNF-α, could help mitigate liver injury by suppressing inflammatory cascades [96]. Additionally, modulating T-cell response to reduce hepatocyte damage while preserving its antiviral immunity is noteworthy. For instance, boosting regulatory T-cell (Treg) populations or enhancing their suppressive functions via Treg therapy can circumvent inflammation mediated by overactive CD8+ T cells without compromising virus clearance. Immune checkpoint inhibitors, such as anti-CTLA-4 and anti-PD-1 antibodies, can restore T-cell function and prevent its exhaustion, leading to an effective antiviral response without extensive liver inflammation and injury [95].

Corticosteroids are valued for their anti-inflammatory properties, yet they must be administered with care to minimize the risk of secondary infections [96]. Clinically, corticosteroids such as dexamethasone significantly reduced mortality in critically ill patients affected by excessive inflammation [96]. Dexamethasone disturbs the activation of NF-κB and AP-1, which subsequently interferes with the transcription of pro- and anti-inflammatory genes, including TNF-α, IL-6, IFN-γ and IL-10 [97]. The interference progressively re-establishes the host immune response, averting cytokine storms while preserving antiviral defenses [102].

### 5.3. Liver Protection Strategies

Protecting hepatocytes from destructive reactive molecules of oxidative stress and mitochondrial dysfunction can help mitigate DENV-2-induced liver damage. This can be achieved through the use of antioxidants such as N-acetylcysteine (NAC), which replenishes intracellular glutathione (GSH) to re-establish cellular redox balance [98]. Besides NAC supplements, uptake of GSH precursors through consumption of garlic, cruciferous vegetables such as broccoli, and whey protein can also help enhance antioxidative capacity and limit inflammation in affected individuals [103].

Likewise, compounds that support mitochondrial repair and protection, including NAD^+^ precursors, can aid in restoring mitochondrial respiration and redox functions in virus-induced liver injury models [99]. For example, CoQ10 aids in restoring mitochondrial membrane integrity and ATP production, and L-carnitine facilitates fatty acid oxidation and energy metabolism, which may help lessen liver damage in viral infections [99,100]. Consequently, mitigating DENV-2–induced liver dysfunction may require a multifaceted strategy that combines viral replication suppression, inflammatory modulation, and protection of hepatocytes from oxidative and metabolic stress—an approach that must be substantiated through well-designed clinical trials in dengue patients.

## 6. Challenges and Future Directions

Despite substantial progress in elucidating the hepatic pathogenesis of DENV-2, several critical challenges remain. Individual variability in immune responses can markedly influence the efficacy of therapeutic strategies, complicating the identification of reliable predictive biomarkers and the design of targeted interventions. Moreover, the currently available experimental models for dengue, ranging from in vitro cell lines to in vivo systems, are limited in their capacity to recapitulate the full complexity of human liver pathology during DENV-2 infection. This underscores the urgent need for more physiologically relevant platforms, such as humanized mouse models, to better simulate virus–host interactions and the intricate immune responses involved. Looking ahead, research should prioritize the development of therapeutic strategies that achieve an optimal balance between efficient viral clearance and the prevention of immune-mediated hepatic injury. Such approaches hold significant potential to improve clinical outcomes and reduce the burden of severe dengue.

## 7. Conclusions

DENV-2 continues to pose a major global health burden, with hepatic involvement remaining a critical determinant of disease severity. However, the precise molecular and immunopathological mechanisms driving liver injury are still incompletely defined. Current models inadequately recapitulate human hepatic responses, limiting mechanistic exploration and translational progress. Future research should emphasize integrative, multi-omic, and model-driven approaches to unravel host–virus interactions, identify early biomarkers of hepatic dysfunction, and develop targeted antiviral and hepatoprotective therapies aimed at mitigating severe dengue outcomes.

## Figures and Tables

**Figure 1 ijms-26-10904-f001:**
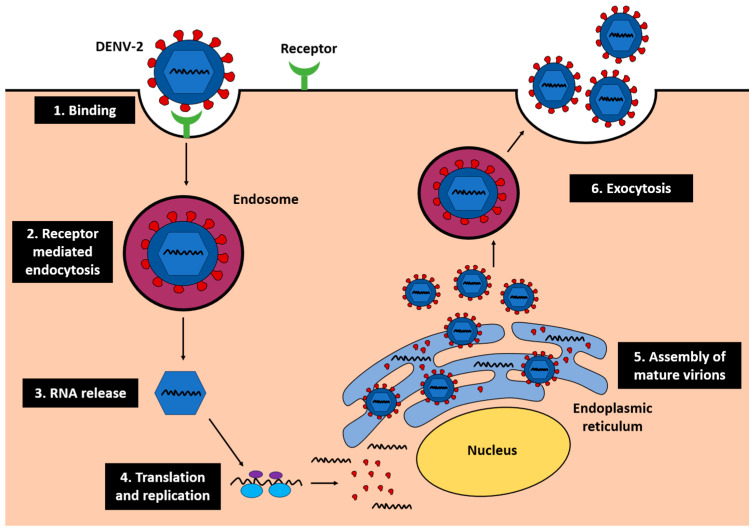
Life cycle of DENV-2 in the liver. Within the ER, translation of viral proteins and assembly of virions contribute to ER stress and dysfunction.

**Figure 2 ijms-26-10904-f002:**
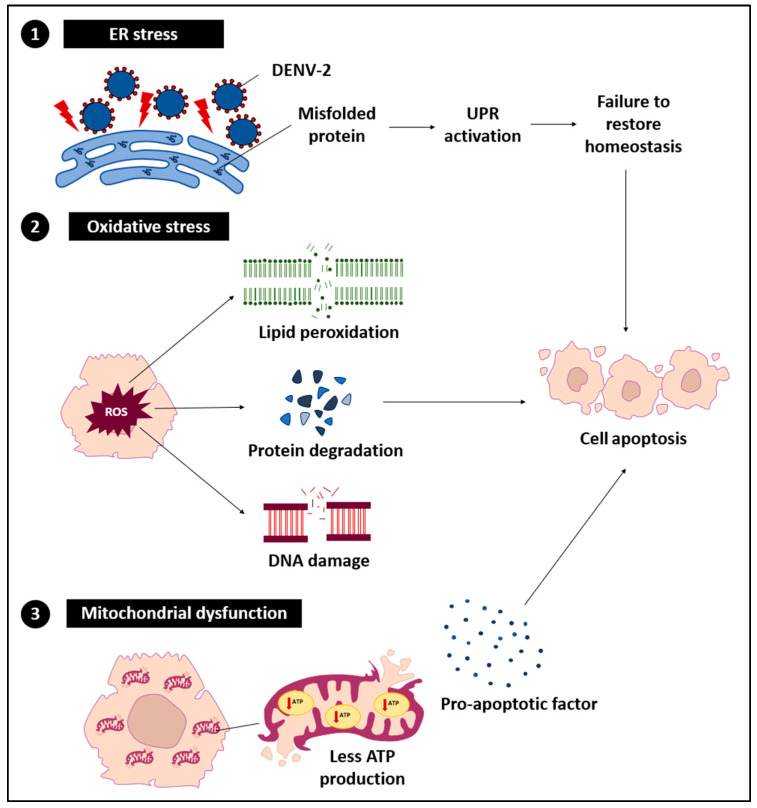
Impacts of DENV-2 infection on hepatocytes, including ER and oxidative stress, and mitochondrial dysfunction.

**Figure 3 ijms-26-10904-f003:**
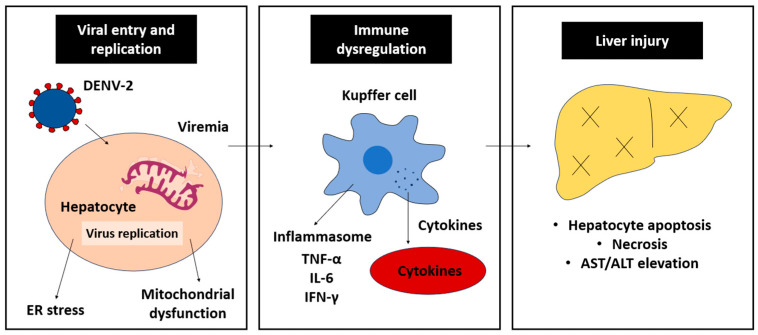
Hepatic pathogenesis of DENV-2. The virus-induced cytokine storm and cellular immune response promote cellular apoptosis and liver injury.

**Table 1 ijms-26-10904-t001:** Key receptors and co-receptors involved in DENV-2 attachment and entry into hepatocytes.

Receptor	Cell Type	Role in DENV-2 Entry	Function Type	Reference
Dendritic cell-specific intercellular adhesion molecule-3-grabbing non-integrin (DC-SIGN, CD209)	Kupffer cells, dendritic cells	Binds to the DENV E protein, enhancing viral attachment and uptake.	Primary receptor	[31]
Mannose Receptor (MR, CD206)	Kupffer cells, endothelial cells	Recognizes viral glycans, aiding in attachment but requiring additional receptors for entry.	Co-receptor	[32]
AXL receptor tyrosine kinase (Tyro3, Axl, and Mer (TAM) receptor family)	Hepatocytes, endothelial cells	Facilitates viral internalization through Gas6-mediated phosphatidylserine recognition, enhancing infection.	Co-receptor	[33]

**Table 2 ijms-26-10904-t002:** Summary of potential therapeutic targets and strategies for DENV-2-induced liver injury.

Therapeutic Category	Target/ Mechanism	Representative Compounds or Strategies	Expected Outcome	References
Antiviral strategies	Non-structural Protein (NS) 3 protease Inhibition	Small-molecule protease inhibitors	Inhibit viral protein processing and replication	[92,93]
	NS5 polymerase inhibition	Nucleoside analogs, polymerase inhibitors	Reduce viral RNA synthesis and load	
	Receptor blockade	Entry inhibitors, receptor antagonists	Prevent viral attachment and entry into hepatocytes	[94]
Immunomodulatory approaches	TNF-α and cytokine suppression	Anti-TNF-α agents, corticosteroids (e.g., dexamethasone)	Reduce cytokine storm and inflammation	[95]
	T-cell regulation	Regulatory T cell (Treg)-based therapy, immune checkpoint inhibitors (anti-programmed cell death protein-1 (PD-1), anti-cytotoxic T-lymphocyte–associated protein-4 (CTLA-4))	Balance immune response, prevent T-cell exhaustion	[96,97]
Hepatoprotective measures	Oxidative stress reduction	N-acetylcysteine (NAC), dietary antioxidants	Replenish GSH, limit oxidative injury	[98]
	Mitochondrial protection	Coenzyme Q10 (CoQ10), L-carnitine, nicotinamide adenine dinucleotide (NAD^+^) precursors	Restore mitochondrial function and energy balance	[99,100]

## Data Availability

Dataset available on request from the authors.

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
