# Peer review of "Mechanistic Insights into Hepatic Pathogenesis of Dengue Virus Serotype-2: Host–Virus Interactions, Immune Dysregulation, and Therapeutic Perspectives"

_ijms, 2025, doi:10.3390/ijms262210904_

Round 1

Reviewer 1 Report

Comments and Suggestions for Authors

Overall Evaluation

The manuscript provides a comprehensive and well-structured review of the hepatic pathogenesis of Dengue Virus Serotype-2 (DENV-2), focusing on molecular mechanisms, host–virus interactions, immune dysregulation, and emerging therapeutic approaches. The topic is timely and relevant given the global burden of dengue and the rising clinical importance of hepatic complications. The review is detailed, logically organized, and supported by appropriate references.

However, several sections would benefit from minor refinement to improve scientific precision, flow, and clarity. The manuscript is suitable for publication after minor revisions as outlined below.

Major Strengths

  1. Comprehensive coverage: The review effectively integrates findings from both in vitro and in vivo studies, offering mechanistic insights into DENV-2–induced liver injury.

  2. Logical structure: The manuscript flows coherently from viral entry mechanisms to immune-mediated damage and therapeutic perspectives.

  3. Current and relevant citations: Most references are recent (2020–2024), reflecting up-to-date knowledge.

  4. Scientific illustrations: The inclusion of schematic diagrams (e.g., Figures 1 and 2) significantly enhances readability and conceptual understanding.

  5. Clinical and translational implications: The discussion linking mechanistic understanding to therapeutic targets is valuable and well aligned with the journal’s scope.

Minor Comments

  1. Abstract and Introduction

    • The abstract is informative but slightly long. Consider condensing repetitive statements (e.g., “highlighting viral entry pathways, endoplasmic reticulum stress, mitochondrial dysfunction, and oxidative damage”) into a concise summary.

    • In the introduction, please clarify the rationale for focusing specifically on DENV-2 rather than all serotypes. A brief comparison (one to two sentences) would strengthen the justification.

  2. Figures and Tables

    • Ensure that all figures (Figures 1–2) and Table 1 are presented with high resolution and proper legends consistent with MDPI formatting guidelines.

    • For Table 1, include reference numbers in the table or as footnotes to facilitate traceability.

  3. Section 3.2 (Cellular Stress Mechanisms)

    • Some mechanistic overlaps between ER stress, oxidative stress, and mitochondrial dysfunction could be synthesized to avoid redundancy.

    • Please ensure that abbreviations (e.g., UPR, ROS) are defined at first mention in the main text.

  4. Language and Style

    • Minor grammatical inconsistencies and typographical errors are present (e.g., “it reduces the viral load and limits liver damage, particularly in the acute phase of DENV infections when the viral replication is most active [91,94]” — missing punctuation before the citation).

    • Consider a light English editing pass to improve sentence smoothness and readability.

  5. Therapeutic Section (Section 5)

    • The section is informative but could benefit from clearer subdivision between antiviral strategies, immunomodulatory approaches, and hepatoprotective measures.

    • If possible, include a concise summary table listing key therapeutic targets, mechanisms, and representative compounds.

  6. Conclusion

    • The conclusion effectively summarizes the main findings but could be strengthened by briefly emphasizing the gaps in current knowledge and how future research might address them.

  7. Formatting

    • Ensure consistency with MDPI’s reference style (e.g., punctuation, author initials, DOI formatting).

    • Verify that all abbreviations are included in the “Abbreviations” section and appear in the same form as in the text.

Final Assessment

This is a scientifically valuable and well-organized review that contributes meaningfully to the understanding of DENV-2 hepatic pathogenesis. With minor textual and structural refinements as suggested, the manuscript will meet the standards for publication in International Journal of Molecular Sciences.

Author Response

  1. Summary

Thank you very much for taking the time to review this manuscript. Please find the detailed responses below and the corresponding revisions/corrections are highlighted in the manuscript.

  1. Point-by-point response to Comments and Suggestions for Authors

Comments 1:

The abstract is informative but slightly long. Consider condensing repetitive statements (e.g., “highlighting viral entry pathways, endoplasmic reticulum stress, mitochondrial dysfunction, and oxidative damage”) into a concise summary.

Response 1:

The abstract has been revised (Lines 14-23).

Comments 2:

In the introduction, please clarify the rationale for focusing specifically on DENV-2 rather than all serotypes. A brief comparison (one to two sentences) would strengthen the justification.

Response 2:

A justification for why DENV-2 is especially discussed in this manuscript is added Lines 34-37).

Comments 3:

Ensure that all figures (Figures 1–2) and Table 1 are presented with high resolution and proper legends consistent with MDPI formatting guidelines.

Response 3:

All figures (Fig. 1-3) are replaced with the high-resolution ones produced in-house using Microsoft Power Point. The legends are formatted according to the MDPI guidelines.

Comments 4:

For Table 1, include reference numbers in the table or as footnotes to facilitate traceability.

Response 4:

A reference column is added to Table 1 for the mentioned purpose (Lines 150-151).

Comments 5:

Some mechanistic overlaps between ER stress, oxidative stress, and mitochondrial dysfunction could be synthesized to avoid redundancy.

Response 5:

Sections 3.2.1, 3.2.2 and 3.2.3 are revised and improved to avoid redundancy in the information (Lines 213-253).

Comments 6:

Please ensure that abbreviations (e.g., UPR, ROS) are defined at first mention in the main text.

Response 6:

All abbreviations used are preceded by their full forms wherever they are first mentioned in the text, e.g. unfolded protein response (UPR) (Line 218) and reactive oxygen species (ROS) (Line 67).

Comments 7:

Minor grammatical inconsistencies and typographical errors are present (e.g., “it reduces the viral load and limits liver damage, particularly in the acute phase of DENV infections when the viral replication is most active [91,94]” — missing punctuation before the citation).

Response 7:

Those lines are revised and improved grammatically (Lines 393-396).

Comments 8:

Consider a light English editing pass to improve sentence smoothness and readability.

Response 8:

The revised manuscript has been proofread by a native English-speaking scientist to improve its quality.

Comments 9:

The section is informative but could benefit from clearer subdivision between antiviral strategies, immunomodulatory approaches, and hepatoprotective measures.

Response 9:

To better present the section, an introductory note (Lines 387-396) is provided prior to the subsections emphasizing antiviral strategies, immunomodulatory approaches, and hepatoprotective measures. The information is also summarized in Table 2 for better readability.

Comments 10:

If possible, include a concise summary table listing key therapeutic targets, mechanisms, and representative compounds.

Response 10:

The information is summarized in Table 2 for better readability (Line 394).

Comments 11:

The conclusion effectively summarizes the main findings but could be strengthened by briefly emphasizing the gaps in current knowledge and how future research might address them.

Response 11:

The conclusion is revised to recapitulate the essence of this review and suggest the way forward for similar research in filling the gaps in the current knowledge and future research (Lines 469-476).

Comments 12:

Ensure consistency with MDPI’s reference style (e.g., punctuation, author initials, DOI formatting).

Response 12:

The manuscript is revised according to the MDPI reference style.

Comments 13:

Verify that all abbreviations are included in the “Abbreviations” section and appear in the same form as in the text.

Response 13:

An Abbreviation section is included at the end of the manuscript (Lines 486-498).

Reviewer 2 Report

Comments and Suggestions for Authors

The review “Mechanistic Insights into Hepatic Pathogenesis of Dengue Virus Serotype-2: Host–Virus Interactions, Immune Dysregulation, and Therapeutic Perspectives” summarize new data on the Dengue virus and its interaction with host cells at the molecular level. Also review provide information about existing therapeutic modalities. It is not clear why the name of review contain "mechanistic".

The introduction is written in sufficient volume, rather succinctly, but allows you to assess the relevance of the consideration of this topic in the review. Nevertheless, it would be possible to give examples of currently existing reviews on this topic, if any.

In line 51-54 “Severe dengue cases, such as DHF and DSS, often involve liver dysfunctions, marked by the elevated ALT and AST levels, hepatomegaly, jaundice, and, in extreme cases, acute liver failure. These biochemical markers not only reflect hepatocellular injury but also correlate with increased disease severity and extended hospital stays” hepatomegaly, jaundice, acute liver failure are not biochemical markers.

Further, there is a lack of minimal information about the life cycle of the virus in liver cells and illustrations to it, allowing for a better understanding of the following sections.

The same information is often repeated in the text itself (lines 39 and 52 as example).

The purpose of the section on cell models for studying the Dengue virus is unclear. This section seems superfluous in this review, based on what the authors stated in the title and abstract.

In the section on existing therapeutic approaches, you need to add some kind of summary table that allows you to somehow compare them with each other.

Taking into account the comments, I recommend the publication after the major revision.

Author Response

  1. Summary

Thank you very much for taking the time to review this manuscript. Please find the detailed responses below and the corresponding revisions/corrections are highlighted in the manuscript.

  1. Point-by-point response to Comments and Suggestions for Authors

Comments 1:

The review “Mechanistic Insights into Hepatic Pathogenesis of Dengue Virus Serotype-2: Host–Virus Interactions, Immune Dysregulation, and Therapeutic Perspectives” summarize new data on the Dengue virus and its interaction with host cells at the molecular level. Also review provide information about existing therapeutic modalities. It is not clear why the name of review contain "mechanistic".

Response 1:

The term “mechanistic” was deliberately included in the title to emphasize the review’s detailed exploration of the molecular and cellular processes underlying DENV-2–induced hepatic pathogenesis. This terminology reflects the manuscript’s focus on elucidating the mechanistic basis of liver involvement, encompassing viral entry and replication, endoplasmic reticulum stress, oxidative injury, and immune dysregulation. To ensure coherence and transparency, the Introduction has been revised to clearly articulate this mechanistic perspective and to align the manuscript’s scope and objectives with its title (Lines 44-50).

Comments 2:

The introduction is written in sufficient volume, rather succinctly, but allows you to assess the relevance of the consideration of this topic in the review. Nevertheless, it would be possible to give examples of currently existing reviews on this topic, if any.

Response 2:

Although limitedly available, three related works (Ref. 6-8) are included in Line 45.

Comments 3:

In line 51-54 “Severe dengue cases, such as DHF and DSS, often involve liver dysfunctions, marked by the elevated ALT and AST levels, hepatomegaly, jaundice, and, in extreme cases, acute liver failure. These biochemical markers not only reflect hepatocellular injury but also correlate with increased disease severity and extended hospital stays” hepatomegaly, jaundice, acute liver failure are not biochemical markers.

Response 3:

Lines 55-57 are revised to better explain the implications of AST and ALT in DENV-2 infection.

Comments 4:

Further, there is a lack of minimal information about the life cycle of the virus in liver cells and illustrations to it, allowing for a better understanding of the following sections.

Response 4:

Life cycle of DENV-2 in liver cells is elaborated in Lines 57-60, and Figure 1 (Line 62) is added to better explain the virus replication.

Comments 5:

The same information is often repeated in the text itself (lines 39 and 52 as example).

Response 5:

The manuscript has been revised thoroughly to avoid redundancy and improve readability.

Comments 6:

The purpose of the section on cell models for studying the Dengue virus is unclear. This section seems superfluous in this review, based on what the authors stated in the title and abstract.

Response 6:

The section on cell models was retained to provide methodological context supporting the mechanistic discussion, as these models form the basis for understanding DENV-2–induced hepatic injury. The section has been revised to clearly link each model to specific mechanistic insights and streamline its relevance to the review’s overall focus. An introductory note is added to briefly highlight the importance of this section in understanding liver pathology in DENV-2 infection (Lines 320-326).

Comments 7:

In the section on existing therapeutic approaches, you need to add some kind of summary table that allows you to somehow compare them with each other.

Response 7:

To better present the section, an introductory note (Lines 387-396) is provided prior to the subsections emphasizing antiviral strategies, immunomodulatory approaches, and hepatoprotective measures. The information is also summarized in Table 2 for better readability.

Round 2

Reviewer 2 Report

Comments and Suggestions for Authors

The authors took into account all the necessary comments and made corrections to the article. The work can be accepted for publishing in its current form.